# SARS-CoV-2 Infection and CMV Dissemination in Transplant Recipients as a Treatment for Chagas Cardiomyopathy: A Case Report

**DOI:** 10.3390/tropicalmed6010022

**Published:** 2021-02-10

**Authors:** Sarah Cristina Gozzi-Silva, Gil Benard, Ricardo Wesley Alberca, Tatiana Mina Yendo, Franciane Mouradian Emidio Teixeira, Luana de Mendonça Oliveira, Danielle Rosa Beserra, Anna Julia Pietrobon, Emily Araujo de Oliveira, Anna Cláudia Calvielli Castelo Branco, Milena Mary de Souza Andrade, Iara Grigoletto Fernandes, Nátalli Zanete Pereira, Yasmim Álefe Leuzzi Ramos, Julia Cataldo Lima, Bruna Provenci, Sandrigo Mangini, Alberto José da Silva Duarte, Maria Notomi Sato

**Affiliations:** 1Laboratory of Dermatology and Immunodeficiencies (LIM-56), Institute of Tropical Medicine of School of Medicine of São Paulo (FMUSP), 05403-000 São Paulo, Brazil; bengil60@gmail.com (G.B.); ricardowesley@usp.br (R.W.A.); tatiana.yendo@gmail.com (T.M.Y.); franciane.mteixeira@usp.br (F.M.E.T.); luana.mendonca@usp.br (L.d.M.O.); daniellerb@usp.br (D.R.B.); pietrobonaj@usp.br (A.J.P.); emilyaraujo@hotmail.com (E.A.d.O.); annabranco@usp.br (A.C.C.C.B.); milena_mary@hotmail.com (M.M.d.S.A.); iaragf@usp.br (I.G.F.); natalli@usp.br (N.Z.P.); yasmim.leuzzi@usp.br (Y.Á.L.R.); juliacataldolima@hotmail.com (J.C.L.); adjsduar@usp.br (A.J.d.S.D.); marisato@usp.br (M.N.S.); 2Institute of Biomedical Sciences, University of São Paulo, 05508-000 São Paulo, Brazil; 3Instituto do Coração (Incor), Hospital das Clínicas, School of Medicine of University of São Paulo (HCFMUSP), 05403-900 São Paulo, Brazil; bruna_provenci@hotmail.com (B.P.); sanmangini@ig.com.br (S.M.)

**Keywords:** SARS-CoV-2, COVID-19, heart transplant, CMV, Chagas disease, infection, severity

## Abstract

Coronavirus disease 2019 (COVID-19) is caused by severe acute respiratory syndrome coronavirus 2 (SARS-CoV-2). COVID-19 has infected over 90 million people worldwide, therefore it is considered a pandemic. SARS-CoV-2 infection can lead to severe pneumonia, acute respiratory distress syndrome (ARDS), septic shock, and/or organ failure. Individuals receiving a heart transplantation (HT) may be at higher risk of adverse outcomes attributable to COVID-19 due to immunosuppressives, as well as concomitant infections that may also influence the prognoses. Herein, we describe the first report of two cases of HT recipients with concomitant infections by SARS-CoV-2, *Trypanosoma cruzi,* and cytomegalovirus (CMV) dissemination, from the first day of hospitalization due to COVID-19 in the intensive care unit (ICU) until the death of the patients.

## 1. Introduction

Coronavirus disease 2019 (COVID-19), an infectious respiratory disease caused by severe acute respiratory syndrome coronavirus 2 (SARS-CoV-2), has spread on a pandemic scale since the first case was reported in Wuhan, China, in 2019 [1]. Most patients with the disease have mild-to-moderate symptoms; however, approximately 15% develop severe pneumonia, while approximately 5% develop acute respiratory distress syndrome (ARDS), septic shock, and/or organ failure [2]. Lymphopenia is a recurrent feature in these patients, with a significant reduction in CD4+ T cells, CD8+ T cells, B cells, and natural killer (NK) cells [3], increasing the susceptibility of patients to severe illness and co-infection [4]. In fact, co-infections are associated with worsening of the clinical condition [5].

Chagas disease (CD) is a zoonosis whose etiologic agent is the protozoan *Trypanosoma cruzi*. It is estimated that 6–8 million people worldwide are infected [6]. CD has two distinct phases: an acute one, which is rare, with a strong production of type 1 cytokines, and a chronic phase, which develops in 30–40% of CD cases. The chronic phase of CD can be characterized by cardiomyopathy, arrhythmias, megaviscera, and, more rarely, polyneuropathy and stroke [7]. TCD4+ and TCD8+ lymphocytes are the main cells responsible for controlling parasitic infection. However, the immune response also contributes to tissue damage and pathology [8].

Chagasic cardiomyopathy represents the main cause of mortality from this disease, which can lead to heart failure, whose indication for treatment, especially in endemic countries, may include heart transplantation (HT) as a strategy to curb the evolution of this complication [6,9]. However, the immunosuppression and possible reactivation of the causative agent *T. cruzi* following HT require intensive clinical care and laboratory monitoring [10].

Other infections such as that caused by cytomegalovirus (CMV), a herpes virus that infects up to 60–100% of people in adulthood, are associated with transplant complications [11]. Primary CMV infection is generally asymptomatic in immunocompetent individuals, with the virus generating a latent infection. On the contrary, in immunocompromised and immunosuppressed populations, such as solid organ transplantation, hematopoietic stem cell transplantation, and HIV/AIDS patients, CMV reactivation is responsible for significant morbidity and mortality [12]. Although reactivation of infections such as CMV and CD are often described in transplant recipients, there are no reports of concomitant infection by *T. cruzi*, CMV, and SARS-CoV-2 in the context of HT in the literature to date. 

Therefore, in this report, we investigated the progress of two patients who underwent HT at the Heart Institute of Hospital das Clínicas (Incor) and were subsequently transferred to the special intensive care unit (ICU) of the Hospital das Clínicas (Hospital das Clínicas, Faculty of Medicine, University of São Paulo-HCFMUSP) due to SARS-CoV-2 infection. These patients were diagnosed with COVID-19 by nasopharyngeal detection of SARS-CoV-2 RNA using reverse transcriptase polymerase chain reaction (RT-PCR). In addition, during hospitalization, CMV dissemination was evidenced by quantitative DNA detection in the blood. We describe herein the laboratory data from the first day of hospitalization due to COVID-19 until the death of the patients. 

The baseline characteristics of these two patients, as well as their past clinical data, are summarized in Table 1. 

## 2. Case Report

### 2.1. Patient 1

Female, 55 years old, presenting positive serology for CD since 2015, received HT on 2 May as a treatment for Chagas cardiomyopathy (Figure 1A) and underwent immunosuppressive therapy with methylprednisolone and azathioprine. Post-transplantation, she developed pneumonia, treated with meropenem and linezolid, and a remittent *Candida tropicalis* infection, treated with micafungin, with improvement. However, on 29 May, she developed dyspnea and desaturation, with a chest tomography suggestive of COVID-19. Her polymerase chain reaction (PCR) for SARS-CoV-2 was positive on 1 June.

On 3 June, she was transferred to the ICU specialized for treatment of severe SARS-CoV-2 infections. Laboratory analyses then showed that the patient had a reduced number of erythrocytes and hemoglobin level, and these reductions became more accentuated by day 33 in the ICU (Figure 2A,B). On this same day, the number of neutrophils peaked (Figure 2D). From the 4th to the 16th day in ICU, important lymphopenia developed (Figure 2F), resulting in an increase in the neutrophil-to-lymphocyte ratio in the same period (Figure 2H). Additionally, from the eighth day onward, she presented severe thrombocytopenia that persisted until death (Figure 2I). 

Throughout the ICU hospitalization period, there were sustained high levels of creatinine, urea, D-dimer, C-reactive protein (CRP), and lactate dehydrogenase (Figure 2J,K,N,S,W). On the 32nd day, she presented a sharp increase in prothrombin time and activated partial thromboplastin time. 

The cardiac function markers of creatine kinase myocardial band (CK-MB) and troponin remained elevated from the first days of ICU admission until death (Figure 2U,X). On the day of admission to the ICU, the N-terminal pro b-type natriuretic peptide (NT pro-BNP) was 70,000 pg/mL (reference value of <125 pg/mL) (Figure 2V). On the third day of hospitalization, NT pro-BNP (155,117 pg/mL) and troponin (0.28 ng/mL, reference value of <0.014 ng/mL) peaked, being related to the period in which she presented signs of graft rejection, which was treated with pulse methylprednisolone, thymoglobulin, and plasmapheresis. In addition, disseminated CMV infection was diagnosed (RT-PCR viral loads of 122 IU/mL on day 10 and 54 IU/mL on day 19 of hospitalization), for which she received ganciclovir. 

After 47 days after diagnosis of SARS-CoV-2, the patient died (18 July) due to multiple organ dysfunctions associated with COVID-19.

### 2.2. Patient 2

Male, 62 years old, previously diagnosed with CD, received HT on 5 December 2019 as treatment for Chagas cardiomyopathy (Figure 1B). He underwent immunosuppressive therapy with cyclosporine, azathioprine, and prednisone. During postoperative hospitalization he presented Chagas reactivation characterized by skin biopsy and humoral graft rejection, being treated with plasmapheresis and methylprednisolone.

The patient was admitted to the ICU of the Hospital das Clínicas on 8 June, presenting cellulitis, deep vein thrombosis, and Chagas reactivation (a lower limb chagoma). The latter was treated with benznidazol. During hospitalization, he tested positive for PCR of SARS-CoV-2 on 29 June. He developed ARDS and septic and cardiogenic shock, which were the causes of his death.

Throughout the ICU hospitalization period, he maintained decreased levels of erythrocytes, hemoglobin, and lymphocytes (Figure 2A,B,F), as well as a high neutrophil/lymphocyte ratio (Figure 2H). From the 12th day, the number of platelets decreased and continued until the time of death (Figure 2I). The values of creatinine, urea, and glucose also remained high throughout the hospitalization period (Figure 2J,K,T). In addition, disseminated CMV (viral loads of 711 IU/mL detected on the 28th day and 1477 IU/mL on the 35th day of hospitalization) was diagnosed and treated with ganciclovir.

One day before death, the following laboratory parameters peaked: D-dimer (7612 ng/mL FEU, reference value of <500 ng/mL FEU), CRP (280.6 ng/mL, reference value of 0.300 ng/mL), CK-MB (8.38 ng/mL, reference value of 0.10–4.94 ng/mL), NT pro-BNP (55,393 pg/mL), lactate dehydrogenase (1161 U/L, reference value of 135–225 U/L), and troponin (0.701 ng/mL) (Figure 2N,S,U–X).

On 20 July, 36 days after admission to the ICU and 21 days after a positive COVID-19 diagnosis, the patient died.

## 3. Discussion

Herein, we described the first report of triple infection (SARS-CoV-2 infection, *T. cruzi* infection, and CMV dissemination) in HT recipients. Patients received HT as a form of treatment for Chagas cardiomyopathy.

We described the laboratory data from the first day of hospitalization in the ICU due to COVID-19 until the time of death. Both patients were admitted to a referral center for treatment for COVID-19 in the metropolitan region of São Paulo, a city in southeastern Brazil. We hypothesize that the triple infection by SARS-CoV-2 and CMV may have been an important cause of death and of the worsening in CD patients with HT. To date, there have been no similar reports of patients presenting these three concomitant infectious diseases and HT receptors.

COVID-19 infection among transplant recipients increases the potential for developing severe illness [13] and may vary between different organ transplants [14,15].

SARS-CoV-2 entry´s receptor angiotensin converting enzyme 2 (ACE2) and transmembrane protease serine 2 (TMPRSS2) are expressed in different tissues, such as the lungs, heart, liver, kidneys, testicles, thyroid, and adipose tissue [16]. Some patients with COVID-19 develop severe disease characterized by respiratory distress syndrome and systemic manifestations. This condition has been associated with the dysregulated release of pro-inflammatory cytokines, termed “cytokine storm,” which may induce multi-organ failure [1,17].

The use of immunosuppressants and post-surgical opportunistic infections can also lead to damage of multisystem organs or even death [18]. There is a therapeutic paradox here, because while insufficient immunosuppression results in graft loss due to rejection, excessive immunosuppression can result in serious infection, including SARS-COV-2 [13], besides contributing to the reactivation of pathogens [19].

Long-term administration of immunosuppressants to solid organ transplant (SOT) receptors to reduce the risk of graft rejection may increase the risk of respiratory infections [20], although there is no clear clinical evidence of increased morbidity/mortality in SARS-CoV-2 infection [21].

Fernandez-Ruiz and collaborators [22] described a cohort of SOT receptors affected by COVID-19, 44% of whom were kidney transplant recipients, 33.3% liver transplant recipients, and 22.2% heart transplant recipients. The lethality rate was 27.8%, suggesting that SARS-CoV-2 infection had a severe course in SOT recipients.

Recently, we described the first patients with CD affected by SARS-CoV-2. The CD patients presented an increase in COVID-19 laboratory hallmarks and a rapid disease progression. Despite the efforts of the health staff, both patients died [3].

HT may be associated with the reactivation of pathogens, as reported in CD [10]. In a Brazilian cohort, the reactivation rate of Chagas disease after heart transplantation was reported to be 38.8% [23]. This fact made HT as a treatment for Chagas cardiomyopathy initially controversial. However, currently, especially in endemic countries, it is the most viable therapeutic option for patients with end-stage heart failure [24]. We hypothesize that, in the cases presented here, HT followed by reactivation of CD conferred an additional risk factor for the worsening of COVID-19.

Transplant recipients are also commonly affected by CMV reactivation, being associated with significant morbidity and mortality [12] that may worsen the infectious condition of COVID-19 [25]. In SOT, risk factors include the use of immunosuppressants for transplantation, advanced age, acute rejection, and other concomitant infections [11], with all these characteristics being present in both patients described herein. In the present report, we observed that immunosuppression may have contributed to the susceptibility to superinfections and more severe clinical manifestations in individuals undergoing HT.

Since overactivated immune responses can be one of the causes of organ damage, the anti-inflammatory effects of immunosuppression can be protective, reducing the cytokine storm related to complications in COVID-19 [26]. In this context, it has been described that immunosuppressive therapy with calcineurin inhibitors in patients with solid organ transplantation or systemic rheumatic diseases promotes a clinical course in SARS-CoV-2 infection, which is generally mild, and with an apparently low risk of superinfection [27]. In addition, immunosuppression has not been evaluated as a risk factor for SARS or MERS [28].

However, in the present case report, we observed that immunosuppression may have contributed to the susceptibility to SARS-CoV-2 infection, the reactivation of pathogens, and more serious clinical manifestations in individuals undergoing HT. Corroborating our findings, it was shown that patients with COVID-19 and cancer, due to their systemic immunosuppressive condition caused by malignancy and anticancer treatments, such as chemotherapy, had an increased risk of SARS-CoV-2 infection and a worse prognosis [29]. It has also been reported that chronic use of corticosteroids prior to SARS-CoV-2 contamination is associated with critical disease outcomes, including a high risk of death [30].

These contradictory observations show that knowledge about the relationship between COVID-19 and the patient’s immune condition is limited. Further studies are needed to elucidate the immune responses and prognosis of COVID-19.

In general, HT in patient 1 proved to be successful for the treatment of Chagas cardiomyopathy, with no reactivation of the pathogen. However, after SARS-CoV-2 infection, she presented graft rejection, treated with methylprednisolone, thymoglobulin, and plasmapheresis. There is a description in eye transplantation that COVID-19 infection can compromise the balance of immunoregulatory responses that allow graft survival, contributing to rejection in individuals infected with SARS-CoV-2 [31]. Thus, this change in the balance of attenuation of the immune response, such as a reduction in the number of regulatory T cells (Tregs) [32], may favor direct cardiotoxic action and multiple organ dysfunction.

During postoperative hospitalization, patient 2 presented Chagas reactivation, evidenced by skin biopsy, indicating that the CD was not completely controlled. In addition, it is important to consider that immunosuppressive therapy can contribute to the reactivation of CD [33]. After HT, the patient presented graft rejection and was treated with plasmapheresis and methylprednisolone. With this, before the SARS-CoV-2 infection, the patient was already weakened and died 21 days after the infection, in relation to patient 1 who died after 47 days of infection.

## 4. Conclusions

This report highlights the first case of an association between COVID-19, CD, and CMV dissemination in HT recipients. The patients had rapid disease progression to death. We believe that HT and the usage of immunosuppressive drugs, as well as immunosuppression generated by concomitant infections, may be an important risk factor for the development of severe COVID-19, especially in endemic areas with underreported CD infection.

## Figures and Tables

**Figure 1 tropicalmed-06-00022-f001:**
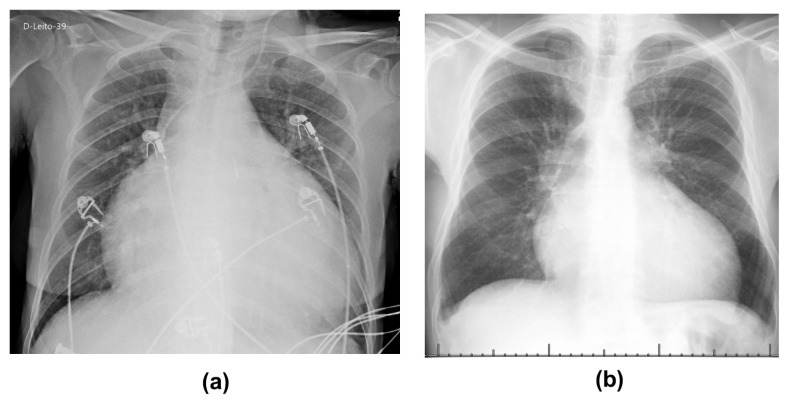
Chest X-ray of (**a**) patient 1 from May 2020 and (**b**) patient 2 from June 2015, taken just before heart transplantation, showing the Chagasic cardiomyopathy.

**Figure 2 tropicalmed-06-00022-f002:**
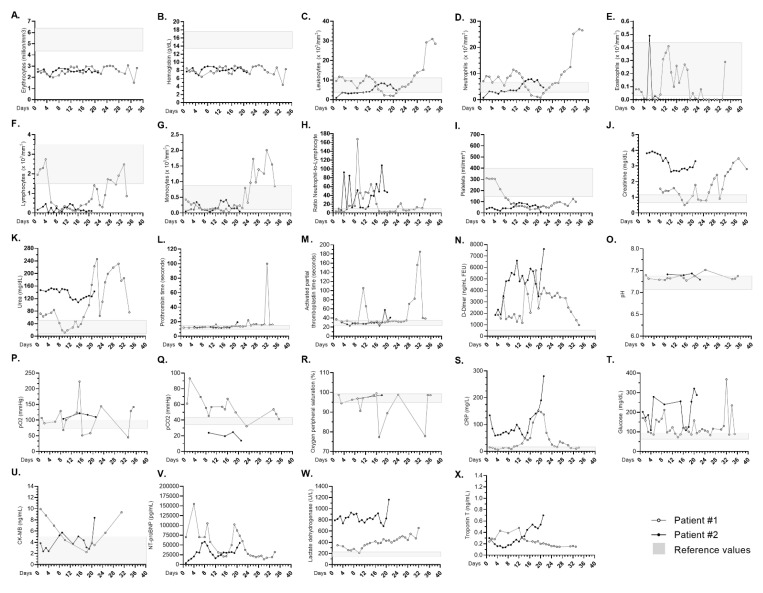
Daily clinical features of patients, from the first day of hospitalization in the ICU until death. Blood levels of (**A**) erythrocytes, (**B**) hemoglobin, (**C**) leukocytes, (**D**) neutrophils, (**E**) eosinophils, (**F**) lymphocytes, (**G**) monocytes, (**H**) neutrophil-to-lymphocyte ratio, (**I**) platelets, (**J**) creatinine, (**K**) urea, (**L**) prothrombin time, (**M**) activated partial thromboplastin time, (**N**) D-dimer, (**O**) pH, (**P**) pO2, (**Q**) pCO2, (**R**) oxygen peripheral, (**S**) CPR, (**T**) glucose, (**U**) CK-MB, (**V**) NT-proBNP, (**W**) lactate dehydrogenase, and (**X**) troponin T. Gray boxes represent the reference values.

**Table 1 tropicalmed-06-00022-t001:** Patients’ characteristics, comorbidities, treatment, and complications.

	Patient 1	Patient 2
**Sex**	Female	Male
**Age**	55	62
**Positive SARS-CoV-2 RT-PCR date**	1 June 2020	29 June 2020
**Heart transplant date**	2 May 2020	5 December 2019
**Death date**	30 days	207 days
**Days between positive SARS-CoV-2 diagnosis and death**	47 days	21 days
**Complications during ICU stay**	ARDS due to COVID–19; heart transplant rejection, disseminated cytomegalovirus; aggravated chronic kidney disease and pressure ulcer	ARDS due to COVID–19, disseminated cytomegalovirus and pancytopenia due to hemophagocytosis
**Previous comorbidities**	Dilated cardiomyopathy of Chagas etiology, hypothyroidism by thyroidectomy by nodule 10 years ago	Cardiomyopathy of Chagas etiology, disseminated cytomegalovirus, deep vein thrombosis, systemic arterial hypertension, diabetes, dyslipidemia, and chronic renal failure
**Previous use of medications**	Carvedilol, losartan, furosemide, levothyroxine, isosorbide, and hydralazine	Unknown
**Medicines used during ICU stay**	Methylprednisolone, azathioprine, anti-thymocyte globulin, cyclosporine, tacrolimus, meropenem, linezolid, micafungin, vancomycin, polymyxin B, tigecycline, amikacin, fluconazole, ganciclovir, and hydrocortisone	Meropenem, colistin, linezolid fluconazole, amikacin, cyclosporine, azathioprine, and prednisone

ARD, acute respiratory distress syndrome; COVID-19, coronavirus disease 2019; ICU, intensive care unit; SARS-CoV-2, severe acute respiratory syndrome coronavirus 2.

## Data Availability

The laboratory data and imaging exams presented herein came from Hospital das Clinics, Faculty of Medicine, University of São Paulo (HCFMUSP) with prior approval for their use.

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
