# Peer review of "SARS-CoV-2 Infection and CMV Dissemination in Transplant Recipients as a Treatment for Chagas Cardiomyopathy: A Case Report"

_tropicalmed, 2021, doi:10.3390/tropicalmed6010022_

Round 1
Reviewer 1 Report
The paper reports two clinical cases of patients who underwent heart transplant for Chagas Disease cardiomyopathy and who died following complications from COVID-19. Concomitant with SARS-CoV-2 infection, the patients tested positive for CMV.
The work itself does not contain particularly original messages and no useful information is deduced for the treatment of this type of patient, but they are certainly two singular clinical cases that have their own intrinsic value at an anecdotal level.
I have some suggestions for the authors to improve the work.
COMMENTS
- I suggest top report the immuno-suppressive therapy in the Table 1.
- Figure legend 1 contains mistakes: (a) is mentioned twice and the date reported for patient 2 is likely wrong since he was transplanted in 2019 and not 2015.
- It is ackward to me that these fragile patients could be infected during a hospital stay. How the patients got infected? Is it known?
- Data concerning the clinical manifestations of SARS-CoV-2 infection in transplanted and immunomodulated patients is currently still debated. Alongside data cited by the authors that seem to indicate greater susceptibility to overinfections and more severe clinical manifestations in transplant recipients, there are others that indicate the opposite. This evidence should also be mentioned in the discussion (e.g. Calcineurin Inhibitor-Based Immunosuppression and COVID-19: Results from a Multidisciplinary Cohort of Patients in Northern Italy - Microorganisms 2020 Jun 30; 8 (7): 977. doi: 10.3390 / microorganisms8070977).
- There are many grammatical and typing errors throughout the text. It would be desirable for a native language to read and correct the text.
Author Response
Dear reviewer,
First of all, I would like to thank you for the observations and suggestions for correction. All of them were very constructive and allowed for a better discussion and clarification of the present case report.
Due to the intense dissemination of SARS-CoV-2 causing interhospital infections of employees, it is considered that the infection of both described patients occurred in the hospital environment, after heart transplant surgery as a treatment for Chagas cardiomyopathy.
As suggested, the anti-inflammatory effects of immunosuppression and its possible protective effect on SARS-CoV-2 infection were discussed. On the other hand, it was emphasizing the different results obtained in the present case report and in other articles cited, addressing immunosuppression as a risk factor for the infection and complications of COVID-19. This discussion was included starting at line 198. In the attached file, there is the case report with the suggested corrections and with the reviewed English.
The caption in Figure 1 was also corrected, however, the 2015 date is correct, we only had access to the radiography of this date.
Immunosuppressive therapy is described in the text and is also present in the list of other medicines.
English review was carried out by MDPI.
If necessary include some more information, please let me know.
Kind regards,
Sarah Gozzi

Reviewer 2 Report
Estimated Authors,
Estimated Editors,
thank you for the opportunity to review this very interesting case report on COVID-19 in patients having received heart transplantation because of Chagas Disease (CD). This article, in my opinion, deserves particular interest for various reasons, and namely:
1) Many people living with CD are particularly vulnerable to the more detrimental consequences of SARS-CoV-2 infection as they are socioeconomically vulnerable and have limited access to healthcare.
2) COVID-19 interacts with the cardiovascular system on multiple levels. SARS-CoV-2 binds to the human angiotensin-converting enzyme 2 (ACE2) receptor mainly expressed in the lungs, heart, and vascular endothelium.
3) There is a concern that COVID-19 disease could potentially trigger reactivation of CD. This potential reactivation could be caused by an acquired hemophagocytic lymphohistiocytosis-like disease (cytokine storm), the virus itself, or even the use of some COVID-19 treatments such as steroids, hydroxychloroquine.
4) Patients having received heart transplantation are immunosuppressed patients, i.e. at increased risk of severe COVID-19, especially those with aggressive underlying disease, active immunosuppressive treatment, or lymphopenia.
The patients described by Gozzi-Silva et al, roughly, can be reported to the aforementioned classifications, representing an illustrative example of how CD and Covid-19 may interact, in the more detrimental way.
Despite the potential interest, in my opinion, the paper from GOZZI-SILVA et al will require some minor but significant improvement before a full publication on TropMed.
More precisely, the two cases (in my opinion, please excuse me and correct me if I'm wrong) are different in this way: while case 1 describes as a substantially successful management of CD in heart transplantion may be complicated by COVID-19 (therefore, we can figure out that SARS-CoV-2 caused the death of the patient by means of a directly cardio-toxic action), in case 2 we are dealing with a CD that was originally not so well controlled (i.e. "he presented Chagas reactivation characterized by skin biopsy"), and was possibly even more complicated to keep under control because of the infection by SARS-CoV-2 and its therapy. In other words, I would suggest the Authors to discuss such possible actions in the final sections of their paper (see https://pubmed.ncbi.nlm.nih.gov/33150134/ for some insights).
Eventually, a minor note: in the text, the acronym HT is not solved at its very first appearance (i.e. heart transplantation).
Author Response
Dear reviewer,
First of all, I would like to thank you for the observations and suggestions for correction. All of them were very constructive and allowed for a better discussion and clarification of the present case report.
In fact, the cases described differ in each other for the reasons you have cited. As suggested, this different COVID-19 progression was discussed in both patients, evidencing the time elapsed between the beginning of the infection and death. This discussion was included starting at line 217. In the attached file, there is the case report with the suggested corrections and with the reviewed English.
The acronym for HT was described in its first appearance, as suggested.
English review was carried out by MDPI.
If necessary include some more information, please let me know.
Kind regards,
Sarah Gozzi
